# MMP-13, VEGF, and Disease Activity in a Cohort of Rheumatoid Arthritis Patients

**DOI:** 10.3390/diagnostics13091653

**Published:** 2023-05-08

**Authors:** Mihail Virgil Boldeanu, Lidia Boldeanu, Oana Mariana Cristea, Dana Alexandra Ciobanu, Sabin Ioan Poenariu, Anda Lorena Dijmărescu, Andreea Lili Bărbulescu, Vlad Pădureanu, Teodor Nicuşor Sas, Ștefan Cristian Dinescu, Florentin Ananu Vreju, Horațiu Valeriu Popoviciu, Răzvan Adrian Ionescu

**Affiliations:** 1Department of Immunology, Faculty of Medicine, University of Medicine and Pharmacy of Craiova, 200349 Craiova, Romania; mihailvirgilboldeanu@gmail.com; 2Department of Microbiology, Faculty of Medicine, University of Medicine and Pharmacy of Craiova, 200349 Craiova, Romania; barulidia@yahoo.com (L.B.); ioneteoana@yahoo.com (O.M.C.); 3Doctoral School, University of Medicine and Pharmacy of Craiova, 200349 Craiova, Romania; ciobanu.dana89@gmail.com (D.A.C.); sabinpoenariu@gmail.com (S.I.P.); 4Department of Obstetrics and Gynecology, Faculty of Medicine, University of Medicine and Pharmacy of Craiova, 200349 Craiova, Romania; lorenadijmarescu@yahoo.com; 5Department of Pharmacology, Faculty of Medicine, University of Medicine and Pharmacy of Craiova, 200349 Craiova, Romania; anbarbulescu@gmail.com; 6Department of Internal Medicine, Faculty of Medicine, University of Medicine and Pharmacy of Craiova, 200349 Craiova, Romania; vldpadureanu@yahoo.com; 7Department of Radiology and Imaging, Faculty of Medicine, University of Medicine and Pharmacy of Craiova, 200349 Craiova, Romania; 8Department of Rheumatology, Faculty of Medicine, University of Medicine and Pharmacy of Craiova, 200349 Craiova, Romania; florin_vreju@yahoo.com; 9Department of Rheumatology, BFK and Medical Rehabilitation, University of Medicine, Pharmacy, Science and Technology of Targu Mures, 540139 Targu Mures, Romania; phoratiu2000@gmail.com; 10Third Internal Medicine Department, ‘Carol Davila’ University of Medicine and Pharmacy, 020021 Bucharest, Romania; tane67@gmail.com

**Keywords:** rheumatoid arthritis, matrix metalloproteinase 13, vascular endothelial growth factor, ultrasonography

## Abstract

Identifying certain serum biomarkers associated with the degree of rheumatoid arthritis (RA) activity can provide us with a more accurate view of the evolution, prognosis, and future quality of life for these patients. Our aim was to analyze the presence and clinical use of matrix metalloproteinase-13 (MMP-13), along with vascular endothelial growth factor (VEGF) and well-known cytokines such as tumor necrosis factor-alpha (TNF-α) and interleukin 6 (IL-6) for patients with RA. We also wanted to identify the possible correlations between MMP-13 and these serological markers, as well as their relationship with disease activity indices, quality of life, and ultrasonographic evaluation. For this purpose, we analyzed serum samples of 34 RA patients and 12 controls. In order to assess serum concentrations for MMP-13, VEGF, TNF-α, and IL-6, we used the enzyme-linked immunosorbent assay (ELISA) technique. Our results concluded that higher levels of MMP-13, VEGF, TNF-α, and IL-6 were present in the serum of RA patients compared to controls, with statistical significance. We furthermore identified moderately positive correlations between VEGF, MMP-13, and disease activity indices, as well as with the ultrasound findings. We also observed that VEGF had the best accuracy (97.80%), for differentiating patients with moderate disease activity. According to the data obtained in our study, that although MMP-13, TNF-α and C-reactive protein (CRP) have the same sensitivity (55.56%), MMP-13 has a better specificity (86.67%) in the diagnosis of patients with DAS28_(4v)_ CRP values corresponding to moderate disease activity. Thus, MMP-13 can be used as a biomarker that can differentiate patients with moderate or low disease activity. VEGF and MMP-13 can be used as additional parameters, along with TNF-α and IL-6, that can provide the clinician a better picture of the inflammatory process, disease activity, and structural damage in patients with RA. Our data can certainly constitute a start point for future research and extended studies with multicenter involvement, to support the selection of individualized and accurate therapeutic management strategies for our patients.

## 1. Introduction

Rheumatoid arthritis (RA) is as a chronic systemic autoimmune disease, which develops through the influence of both genetic and environmental factors and carries a major risk for articular destructive lesions and subsequent functional disability in absence of specific and targeted treatment [1,2,3]. It is a pathology defined by synovial inflammation and hyperplasia, antibody production, destructive changes in cartilage and bone, along with systemic features [4]. 

The process of angiogenesis, with a mandatory role for tissular growth, has a central function in the pathogenesis of several inflammatory autoimmune pathologies, including RA [5,6,7,8,9]. Angiogenesis causes the expansion of synovial tissue in RA, as preexisting vessels facilitate the influx of leukocytes into the synovial space, where they exacerbate the inflammation. The process begins with an increase of specific growth factors, such as vascular endothelial growth factor (VEGF) and fibroblast growth factor (FGF), hypoxia-inducible factors (HIF) that bind to their related receptors on endothelial cells, followed by their activation and release of proteolytic enzymes [10].

VEGF is the specific angiogenic factor for the migration and proliferation of endothelial cells [11,12] and an essential factor involved in the process of synovial proliferation and neovascularization during the development and progression of RA [13,14,15]. Kim et al. observed that VEGF has an important role in the differentiation of osteoclasts, involved in the process of bone erosion, another crucial characteristic of RA [16]. VEGF is expressed at the highest concentration in various rheumatic diseases, including RA, systemic lupus erythematosus, antiphospholipid syndrome, and mixed connective tissue disease [17]. Some studies have reported that serum VEGF concentration increases and correlates with disease activity, C-reactive protein (CRP) level and radiographic progression [18,19], or VEGF has been used as a marker to assess treatment response [20]. Recently, a study reported that serum VEGF is more valuable than traditional factors such as CRP in determining the treatment response of patients receiving biologic disease-modifying antirheumatic drug (bDMARDs) [21]. More recent studies have investigated in an experimental RA model the effect of anti-VEGF therapy (ranibizumab, anti-VEGF antibody; ramucirumab, monoclonal antibody against VEGFR2), in monotherapy or in combination with current therapy (methotrexate or tocilizumab treatment), and the authors found that anti-VEGF reduced bone and cartilage destruction [22,23].

Another major mechanism of cartilage destruction in RA is the breakdown of proteoglycan, such as aggrecan and collagen, in the extracellular matrix (ECM) [24,25,26]. Matrix metalloproteinases (MMPs), and particularly MMP-13, are a family of proteinases that largely contribute to ECM disruption [26,27,28]. Currently, MMP-13 is considered to be the major mediator of this process with subsequent inflammation, observed in osteoarthritis (OA) [29,30,31,32]. As reported in the study by Little et al., MMP-13-deficient mice were observed to be resistant to collagen and aggrecan disruption, a phenomenon that prevents cartilage erosion [31]. This proteinase has been reported to be induced by the pro-inflammatory cytokines, including interleukin (IL)-1β and tumor necrosis factor-alpha (TNF-α) in the articular space [32,33]. It has been observed that in OA and neoplasia, VEGF also increases the expression of several types of MMPs, especially MMP-13 [34,35].

Ultrasonography (US) assessment is a sensitive and reproducible imaging technique that has emerged as a valuable method for daily practice to obtain an accurate evaluation of synovitis, the hallmark of the disease [36,37]. US can inform the examiner on the presence of both acute inflammatory lesions and chronic structural changes in the form of bone erosions. Remission, the endpoint therapeutic purpose, can be evaluated using various composite scores, which cannot properly evaluate underlying structural damage. US examination is the best tool that can accurately assess real remission [38]. Due to several technical advances, US examination has an important role in the quantification of synovitis, and can provide important information for the diagnosis, monitoring, and management of RA [39]. Using this imagistic method as an extension to clinical and biological approaches can help us not only for diagnostic purposes, but for assessing disease activity.

In choosing the research theme, we started from the finding that there is a lack of data regarding MMP-13 involvement in RA, as well as the presumable association between MMP-13 and VEGF. The main focus is identifying whether these biomarkers can help us to improve the real-time assessment of our patients and provide possible input on an individualized therapeutic approach. 

Identifying certain serum biomarkers that are associated with disease activity can provide us a more accurate view regarding evolution, prognosis, and future quality of life for these patients [40,41,42]. 

We also wanted to identify the possible correlations between MMP-13 and VEGF and well-known cytokines such as TNF-α and IL-6, their relationship with disease activity score, and ultrasonographic examination. A final objective was to evaluate the diagnostic accuracy of MMP-13 and VEGF serum levels, using receiver–operator characteristic (ROC) curve analysis.

## 2. Materials and Methods

### 2.1. Subjects and Clinical Assessment

This study included 41 consecutive patients diagnosed with RA, according to the American College of Rheumatology/European League Against Rheumatism (ACR/EULAR) 2010 classification criteria [43], based on history, clinical examination, serological parameters, and imaging investigations performed in the Rheumatology Clinic, Emergency County Clinical Hospital of Craiova. 

Inclusion criteria: definite diagnosis of RA according to ACR criteria; patients with active RA, presenting at least five or more joints with active synovitis (painful and swollen joints), plus two of the following three criteria: morning swelling over 60 min, ESR > 28 mm per hour, and CRP > 20 mg/L, determined quantitatively. Presence of metabolic disorders or other autoimmune conditions represented exclusion criteria from the study.

For the comparative analysis, we also included a control group (C) of 12 healthy subjects, without current or past diagnosis of RA, age-compatible with the RA group. 

For each patient, the initial evaluation form included contact information, demo-graphic data, personal pathological history, clinical manifestations, laboratory tests, disease activity score, complications (anemia, infections, pulmonary involvement, etc.), classes of past RA medication, and current therapy. 

As methods for assessing disease activity, we used the conventional Disease Activity Score using 28 joint counts, with 4 variables, using CRP (DAS28_(4v)_ CRP) [44]. DAS28_(4v)_ CRP values define remission (DAS28_(4v)_ CRP < 2.6), low disease activity (2.6 < DAS28_(4v)_ CRP ≤ 3.2), moderate disease activity (3.2 < DAS28_(4v)_ CRP ≤ 5.1), and high disease activity (5.1 < DAS28_(4v)_ CRP) [45]. According to DAS28_(4v)_ CRP score, we identified 15 patients with low disease activity, and 19 patients with DAS28_(4v)_ CRP values corresponding to moderate disease activity. We excluded from the statistical analysis 3 patients with high disease activity, along with 4 patients that achieved remission. Remission was established according to EULAR criteria and certified by US examination that showed no degree of disease activity (Figure 1). As one main objective was to evaluate the possible relationship between disease activity and MMP-13, VEGF, TNF-α, and IL-6, we considered the exclusion of subjects in remission mandatory. The final evaluated group of RA patients consisted of 34 subjects, divided into 2 subgroups: moderate (M) with a DAS28_(4v)_ CRP score corresponding to moderate disease activity; and low (L), with a DAS28_(4v)_ CRP score corresponding to low disease activity.

Functional status was evaluated using the Health Assessment Questionnaire (HAQ) [46]. We also used the Simplified Disease Activity Index (SDAI), an easy-to-use scoring system that focuses on the main symptoms of RA and has been validated for both research and clinical use. The SDAI score helps clinicians determine the severity of RA based on clinical and laboratory data, being the sum of the following five components: swollen joint count, painful joint count, CRP in mg/dL, and the global assessment of disease activity, assessed both by the patient and by the clinician [47].

The RA group received synthetic disease-modifying antirheumatic drugs (csDMARDs)—methotrexate or leflunomide, related to bDMARDs—adalimumab, etanercept, tocilizumab, certolizumab, rituximab or targeted synthetic drugs (tsDMARDs), represented in our study only by baricitinib. At the moment of sample collection and US assessment all the subjects were receiving TNF-α inhibitors, followed by treatment maintenance or switch in case of non-responder status.

All RA patients and controls signed informed consent before inclusion in the study, after being previously informed about the objectives and stages of the study.

### 2.2. Sample Collection

The biological material collected from patients in both groups (RA group and control group) was blood (approximately 5 mL of venous blood) collected into tubes without additives (Becton Dickinson vacutainer). The clot was separated by centrifugation (3.000× *g* for 10 min) no later than 4 h after harvesting, according to the standard procedure. The serum sample tubes were coded for each patient, sealed to avoid contamination, and stored at temperatures between −20 °C and even −80 °C so that the samples could be processed over a longer period. Before working the patient samples, the frozen samples were left at room temperature to accommodate, and freezing–unfreezing cycle operations were avoided. 

### 2.3. Immunological Investigations

Determination of the serological profile of patients with RA was performed in the Immunology Laboratory of the University of Medicine and Pharmacy of Craiova. For the quantitative assessment of serum concentrations of MMP-13, VEGF, TNF-α, and IL-6, the enzyme-linked immunosorbent assay (ELISA) technique was used. 

We used commercial test: VEGF (Catalog# KHG0111; assay range: 23.8–1500 pg/mL; analytical sensitivity: <5 pg/mL), MMP-13 (Catalog# EHMMP13; assay range: 8.23–6000 pg/mL; analytical sensitivity: 6 pg/mL), TNF-α (Catalog# BMS223-4; assay range: 7.8–500 pg/mL; analytical sensitivity: 2.3 pg/mL; Control Low Lot#185277000, range: 20–100 pg/mL; Control High Lot#185278000, range: 350–700 pg/mL), IL-6 (Catalog# BMS213–2; assay range: 1.56–100 pg/mL; analytical sensitivity: 0.92 pg/mL; Control Low Lot#229983–000, range: 3–10 pg/mL; Control High Lot#229984–000, range: 50–150 pg/mL), Invitrogen, Thermo Fisher Scientific, Inc. (Waltham, MA, USA). 

After thawing, each sample was diluted; the dilutions and working procedure were followed according to the manufacturer’s instructions and the prescribed method. A standard optical analyzer at a wavelength of 450 nm was used during the process.

### 2.4. Ultrasound Examination

US examination was performed with ESAOTE MyLab X8 (EsaoteSpA, Genoa, Italy), using a high-frequency 15 MHz linear array transducer with a power Doppler (PD) unit. The US study was performed by an experienced sonographer, unaware of the clinical and paraclinical laboratory data of the patients. PD was used with the following settings: a pulse repetition frequency (PRF) of 0.75 KHz, gain 18–30 dB, low filter. 

The US examination used the US 7 joints score, proposed by Backhaus et al., which includes the evaluation of the most clinically involved joints of the hand and foot: wrist, metacarpophalangeal (MCP) II and III, proximal interphalangeal (PIP) II and III, and metatarsophalangeal (MTP) II and V. These joints were evaluated for synovitis, tenosynovitis, and superficial bone erosions. Synovitis was scored on a 0–3 scale in both grayscale (GS) and PD ultrasound modes, while erosions and tenosynovitis were recorded as present (1) or absent (0). The semiquantitative assessment of synovitis was based on the EULAR/outcome measures in rheumatology (OMERACT) scoring system [48]. GS synovitis was scored as follows: (0) None = no synovial hypertrophy (SH) or effusion; (1) Minimal = SH with or without effusion up to the level of bone surfaces; (2) Moderate = SH with or without effusion extending beyond the joint line; (3) Severe = SH with or without effusion extending beyond the joint line with convex surface and distension of joint capsule. PD synovitis was scored as follows: (0) None = no Doppler signal; (1) Minimal = up to 3 single Doppler spots or one confluent spot; (2) Moderate = greater than grade 1, but <50% of joint displays Doppler signal; (3) Severe = greater than grade 2, with >50% of the joint showing Doppler signal. The scoring range was 0–27 for GS synovitis, 0–39 for PD synovitis, 0–7 for GS tenosynovitis, 0–21 for PD tenosynovitis, and 0–14 for erosions [49,50].

### 2.5. Ethical Issue

All the steps of this study were conducted in accordance with the standards of ethics established by the institutional committees responsible for monitoring human studies and in compliance with the Helsinki Declaration of 1975, which was revised in 2008. Our study obtained the approval from the Ethics, Academic and Scientific Deontology Committee of the University of Medicine and Pharmacy of Craiova No.136/17.09.2021.

### 2.6. Statistical Analysis

Patients’ data obtained from medical documents were managed and processed with Microsoft Excel. Statistical analysis of the data was performed using the GraphPad Prism 5 trial version (San Diego, CA, USA). Furthermore, data normality was tested by using the Shapiro–Wilk test. Normal variables are presented as mean value with standard deviation (SD). MMP-13 and HAQ were found to be non-normally distributed and are presented as median (interquartile range). The difference between the two groups was estimated by independent *t* test and the Mann–Whitney *U* test for normally distributed and skewed data, respectively. All tests were two-sided and *p* values ≤ 0.05 were considered significant. The existence of correlations between the investigated markers, VEGF, MMP-13, TNF-α, and IL-6, as well as the correlations between serological markers, activity indices, and US parameters [DAS28_(4v)_ (CRP), HAQ, SDAI, power Doppler ultrasound (PDUS), and Erosion scores], were assessed using Spearman’s coefficients (−1 < rho < 1), and are visually presented with the correlation heatmap matrix (color range from bright blue for strong positive correlations to bright red for strong negative correlations). *p*-value < 0.05 was set as statistically significant. To determine the diagnostic performance accuracy of the analyzed serological markers, we used the ROC curve to detect possible threshold values (cut-off), which were used in practice to make the best possible differentiation of patients with moderate forms of RA. We calculated for the various cut-off investigated markers, their sensitivity (Sn), specificity (Sp), Youden index (Sn + Sp −1), as well as the false-positive likelihood ratio [LR(+)] and the false-negative LR(−). Performance is expressed as the area under the ROC curve (AUC), accompanied by the 95% confidence interval (CI) and the statistical *p* for the difference between the calculated AUC and the AUC = 0.05 (marker without discriminative power). The power analysis for our study was performed using G*Power 3.1.9.7, at a 95% confidence level and power factor of 80% for each of the groups. A two-sided *p*-value smaller than 0.05 was considered to be statistically significant. The power test was performed assuming an alpha level of 0.05, the patient data from the moderate and low disease activity groups yielded power between 68% and 84% for the different analyzes.

## 3. Results

### 3.1. Demographics and Clinical Characteristics of the Study Subjects

Our analysis included 34 patients (L and M groups) diagnosed with RA, all of them females, and 12 females as control (C). Regarding age, expressed as mean ± SD, the calculated value for the L group with low disease activity was 52.14 ± 10.20 years, while that for the ones with moderate RA was 54.70 ± 10.30. There was no significant difference in age between the two RA groups (L vs. M) (*p* = 0.356). Also, there was no statistically significant difference between the two groups based on residence (urban/rural).

The mean calculated DAS28_(4v)_ CRP was 2.82 ± 0.21, with limits of 2.61 and 5.00. For SDAI, we reckoned a mean of 6.06 ± 4.33. The HAQ ranged between 0.10 and 1.10, with a calculated mean of 0.42 ± 0.21. There were no statistically significant differences between HAQ values among the two groups (*p* = 0.357). Demographic and clinical characteristics of the study subjects are presented in Table 1.

US examination was used to evaluate all patients, both in GS and PDUS modes (Figure 2). Analyzing US findings (Table 1), we found statistically significant differences between the two groups (L vs. M): PDUS score (9.60 ± 2.72 vs. 17.33 ± 1.64, *p* < 0.0001), Erosion score (2.00 ± 0.85 vs. 4.67 ± 1.64, *p* < 0.0001).

### 3.2. Concentrations of Serum MMP-13, VEGF, TNF and IL-6 Levels and Correlations with Disease Activity Scores

Our results showed statistically significant higher concentrations of MMP-13, VEGF, TNF-α, and IL-6 in the RA compared with the C group (Table 2). Comparing the mean concentrations of serum biomarkers, for M vs. C group, we obtained the following data: MMP-13 (882.00 vs. 402.00 pg/mL, *p* = 0.023), VEGF (1112.00 ± 269.00 vs. 386.00 ± 62.20 pg/mL, *p* < 0.0001), TNF-α (24.20 ± 5.25 vs. 14.20 ± 3.20 pg/mL, *p* < 0.0001), IL-6 (27.40 ± 5.38 vs. 15.90 ± 3.86 pg/mL, *p* < 0.0001). 

Regarding L vs. C group, we obtained the following results: MMP-13 (413.00 vs. 402.00 pg/mL, *p* = 0.625), VEGF (590.00 ± 136.00 vs. 386.00 ± 62.20 pg/mL, *p* = 0.0002), TNF-α (20.30 ± 3.26 vs. 14.20 ± 3.20 pg/mL, *p* = 0.0005), IL-6 (22.70 ± 8.49 vs. 15.90 ± 3.86 pg/mL, *p* = 0.026).

Serum concentrations of VEGF were found to be increased directly related to RA activity, with higher titers for patients with moderate disease activity. There were statistically significant differences between the serum concentrations of VEGF in M vs. L group (*p* < 0.0001). As for MMP-13, we also observed higher values directly related to disease activity: M vs. L group (*p* = 0.041) (Figure 3). 

When analyzing TNF-α and IL-6 results, we detected the same direct proportional relationship to disease activity, as well as statistically significant differences between M vs. L group (for TNF-α, *p* = 0.045 and *p* = 0.038 for IL-6) (Figure 4).

Another valuable point of our analysis is the study of the possible correlations between DAS28_(4v)_ CRP and the serum concentrations of MMP-13, VEGF, TNF-α, and IL-6. 

For the L group, Spearman’s correlation analysis revealed statistically significant positive correlation between DAS28_(4v)_ CRP and the serum concentrations of MMP-13 (rho = 0.580, *p* = 0.023), observations also stated for the M group (rho = 0.530, *p* = 0.024). The results are described in Table 3. 

Regarding SDAI, in the M group, there was a positive direct correlation with the serum concentrations of MMP-13, with statistical significance (rho = 0.559, *p* = 0.016).

### 3.3. Correlations between Serum Concentrations of MMP-13, VEGF, TNF-α, IL-6 and Ultrasonographic Parameters

In L group, the serum concentrations of MMP-13 were found to be positively correlated with the Erosion score, with moderate statistical significance (rho = 0.453, 𝑝 = 0.038). A moderate statistically significant positive correlation between the serum concentration of VEGF and the PDUS score was observed in the M group (rho = 0.500, *p* = 0.035). Regarding Erosion score, we identified no correlation with VEGF. The data are presented in Figure 5 and Figure 6.

### 3.4. Correlations between the Serum Concentrations of MMP-13, VEGF, TNF-α, IL-6, and Inflammatory Markers

There were strong statistically significant positive correlations in the L group between the serum levels of TNF-α and IL-6 and the CRP value (rho = 0. 583, *p* = 0.011 and rho = 0.545, *p* = 0.019) and a weak positive correlation between TNF-α and ESR (rho = 0. 380, *p* = 0.041) (Figure 5). 

In the M group, the serum levels of MMP-13 were moderately positively correlated with ESR (rho = 0.503, *p* = 0.034) and CRP (rho = 0.427, *p* = 0.017). We also registered statistically significant positive correlations between the serum concentrations of VEGF and ESR (moderate correlation, rho = 0.485, *p* = 0.037) and CRP (strong correlation, rho = 0.760, *p* = 0.023) (Figure 6). Other strong statistically significant positive correlations were observed between the serum levels of TNF-α and IL-6 (rho = 0.575, *p* = 0.013), TNF-α and ESR (rho = 0.621, *p* = 0.045), TNF-α and CRP (rho = 0.552, *p* = 0.032), and moderate positive correlation between the serum levels of IL-6 and CRP (rho = 0.371, *p* = 0.028). The data are presented in Figure 6.

### 3.5. Diagnostic Accuracy of the Biomarkers

A final objective of this study was to determine and compare Sn and Sp of MMP-13, VEGF, TNF-α, IL-6, with CRP, PDUS, and DAS28_(4v)_ CRP, along with LR (+) and LR (−), which could provide a precise differentiation for patients with moderate disease activity. By comparing the ROC curves for the seven parameters analyzed (Table 4), we can observe that the best differentiation of patients with moderate disease activity of RA can be obtained using PDUS (100.00% accuracy), followed by VEGF (97.80% accuracy), MMP-13 (71.50% accuracy), and IL-6 (71.10% accuracy). Therefore, we refer to the diagnostic performance of the PDUS measurements when we analyze the predictive utility of the markers investigated (Figure 7A–E).

The AUC for VEGF was 0.978 with a 95% CI of 0.938–1.018 (*p* < 0.0001), hence we can conclude that the AUC value differs by as much as 50%. Moreover, by using the ROC curve for VEGF we can determine a threshold value (926.10 pg/mL) to distinguish between low from moderate disease activity. VEGF had 100.00% Sp, a Sn equal to 77.78%, LR (+) = 11.67, LR (−) = 1.07, and the Youden index was 0.778. 

For MMP-13, we obtained a lower accuracy of 71.50% (703.10 pg/mL threshold, 95% CI: 0.534–0.888, *p* = 0.039, 86.67% Sp, a Sn equal to 55.56%), almost the same as that for IL-6 of 71.10% (24.17 pg/mL threshold, 95% CI: 0.523–0.906, *p* = 0.036, 73.33% Sp, a Sn equal to 72.22%), and a diagnostic accuracy equal to 70.70% for TNF-α (23.36 pg/mL threshold, 95% CI: 0.531–0.884, *p* = 0.043, 80.00% Sp, a Sn equal to 55.56%).

In our study we found that although MMP-13, TNF-α, and CRP had the same Sn (55.56%), MMP-13 had the better Sp (86.67%) for distinguishing patients with moderate disease activity.

## 4. Discussion

Our study continues the research of several published papers that focused on analyzing the role of individual pro-inflammatory cytokines such as TNF-α, IL-6, IL-13, IL-17, or MMP-9 in the development of RA [51,52,53]. 

We analyzed possible relationship between MMP-13 and VEGF, along with evaluating the potential correlations between these biomarkers, disease activity, US examination, and inflammatory markers. 

Analyzing our data, we concluded that higher serum levels of MMP-13 and VEGF were present in RA patients compared with controls. Also, we identified moderately positive correlations between MMP-13, VEGF, and disease activity score. 

Higher serum and synovial fluid levels of VEGF and its receptors, along with the correlations between its levels and disease activity, suggest that VEGF can be additionally used for a better view of disease activity, treatment response, future erosive patterns, and extra-articular involvement in RA patients [54,55,56,57]. 

A study performed by Clavel et al., investigated various angiogenesis markers, VEGF, angiopoietin-1 (Ang-1), and soluble Fms-like tyrosine kinase-1 (sFlt-1) in patients with very early arthritis, in order to assess their relevance for predicting subsequent joint destruction. The authors correlated the levels of these specific angiogenesis markers, determined at baseline and after 1 year, by analyzing the inflammatory process, from both a clinical and biological point of view. They reported that only baseline VEGF levels were correlated with the disease activity score assessed after one year [58]. Moreover, Wojdasiewicz et al. [17], analyzed patients with several rheumatic diseases and found significant differences for serum levels of VEGF in patients with RA, compared with antiphospholipid syndrome and systemic lupus erythematosus subjects. 

In 2020, Kim et al. assessed VEGF, placenta growth factor (PlGF), sFlt-1, and IL-6 levels in the synovial fluid and/or serum of patients with RA, and analyzed their possible correlations with US assessment of synovitis, as well as with therapeutic responses to csDMARDs versus bDMARDs. The data revealed that PlGF and VEGF, major pro-angiogenic factors secreted by synoviocytes, were increased both in synovial fluid and in the serum of RA patients. Their expression was directly related to the severity of synovitis, and serum levels of VEGF correlated both with disease activity and treatment response [21]. 

In agreement with these studies, our analysis found statistically significant differences of serum levels of VEGF depending on DAS28_(4v)_ CRP score, with the highest titers registered for the moderate disease activity group. 

A meta-analysis published in 2018 by Lee et al., with 13 studies including 2508 patients with RA and 2489 controls, revealed significantly higher circulating VEGF levels in patients with RA and a positive correlation between VEGF levels and disease activity in RA [19].

One very important correlation of our study, which confirms other reports, was registered comparing serum levels of VEGF and PDUS score, for patients with a moderate disease activity. However, we did not establish a significant correlation between serum levels of VEGF and the Erosion score. US, an imaging technique with several advantages that make it almost indispensable in current daily practice, has a well-established role not only for diagnostic purposes but also to accurately evaluate the degree of vascularization in certain areas of interest, using PDUS [21,57,59,60,61,62,63,64,65,66,67]. Both grayscale ultrasound (GSUS) and PDUS are useful in identifying patients with clinical remission who still display US subclinical inflammation and others with no sign of US active synovitis, and for establishing an evaluation algorithm that integrates several outcome measures. 

In 2017, Algergawy et al. reported a statistically significant correlation between VEGF synovial level and synovial thickness and/or effusion. They also noted a directly proportional relationship between higher synovial levels of VEGF and PDUS findings, results consequent to endothelial proliferation and synovial neoangiogenesis, mediated throughout VEGF [60].

In another important clinical study performed on a group of 50 patients with early RA with moderate or high disease activity, Misra et al. analyzed the relationship between PDUS features and angiogenic and inflammatory markers. They also aimed to assess the possible correlations between PDUS score and DAS28 CRP, as well as those between disease activity score and both inflammatory and angiogenesis markers. They established that high serum levels of angiogenic and inflammatory markers were present in patients with grade 3 PDUS findings, despite a moderate disease activity score, significantly different to levels determined for the group with a DAS28 CRP corresponding to high disease activity but with lower PDUS score [61]. 

Furthermore, Kim et al. demonstrated that PlGF and VEGF levels were directly related to synovitis, assessed both by GSUS and PDUS, along with inflammatory markers and DAS28 CRP score [21]. 

Our data showed a correlation between VEGF serum levels and inflammatory markers, ESR, and CRP in patients with moderate disease activity, being consistent with previously reported studies. 

MMP-13, the major type II collagen-degrading collagenase, is known to be involved in degradation of the ECM by the destruction of proteoglycan, subsequently leading to the erosion of the articular cartilage. Chondrocytes naturally undergo terminal differentiation towards hypertrophy at growth plates and by a process termed endochondral ossification make essential contributions to normal bone growth. Hypertrophic chondrocytes are characterized by their increased size and expression of multiple phenotypic markers including collagen X, MMP-13, VEGF, and, importantly, decreased expression of type II collagen which is most abundant in articular cartilage [29]. Several studies have shown that chondrocyte mRNA expression of MMP-13, but not MMP-1, -8, or -14, is increased in late-stage human OA cartilage in association with cartilage erosion [68,69,70,71,72]. 

In choosing the research topic, we started from the observation that most of the studies reported to date have investigated the input of MMP-13 in OA, and there have been few papers relating to RA. 

In 1999, Konttinen et al. performed a comparative analysis regarding the expression of 16 different MMPs (MMP-1 to MMP-20) in the synovial membrane of patients with trauma and RA. They showed that MMP-13 an enzyme produced by chondrocytes, present in 9 out of 10 samples of RA patients [26]. In 2020, in a study that included 34 patients with RA and 41 patients with OA, Wu Y et al. reported increased levels of both VEGF and MMP-13 in synovial fluid and tissue of patients with RA, significantly higher than in patients with OA, as well as a positive correlation between VEGF and MMP-13 [73]. 

Our results revealed high serum levels of MMP-13 directly correlated with disease activity score. We also registered statistically significant differences in MMP-13 values for patients with moderate disease activity, when compared with controls, an observation that did not apply to subjects with DAS28_(4v)_ CRP scores corresponding to low disease activity. When analyzing the relationship between MMP-13 and VEGF, a significant result was observed only in the moderate disease activity group. The observation made in our study is partially consistent with that in the study mentioned above. 

As showed by our results, when analyzing the predictive utility of the markers investigated we referred to the diagnostic performance of the PDUS measurements. Comparing the ROC curves for the seven parameters analyzed (MMP-13, VEGF, TNF-α, IL-6, CRP, PDUS, DAS28_(4v)_ CRP), we observed that the best distinction between patients with moderate and low disease activity can be achieved using PDUS, followed by VEGF, MMP-13, and IL-6. 

For VEGF we determined a threshold value of 926.10 pg/mL to separate the two forms of RA, low and moderate disease activity. We observed that VEGF was associated with higher accuracy than the other examined serological markers, at 97.80%. 

To our knowledge, this is the first study to evaluate the predictive utility of MMP-13. According to our results, although MMP-13, TNF-α, and CRP had the same Sn (55.56%), MMP-13 had a better Sp (86.67%) for patients with moderate disease activity.

One important study, published by Hirata et al. in 2013, had the objective of evaluating a multi-biomarker disease activity score as a tool to guide management of RA patients. The study included 125 patients with early RA (with duration of symptoms under 2 years), with a mean duration of symptoms of 43.8 weeks. The proposed score was proved to be related to disease activity, and in the majority of patients it provided a similar result to the clinical indices [74]. Our study included exclusively patients with established RA, undergoing biologic therapy, with a disease duration of minimum 6 years. We aimed to establish the possible input and value of certain immunologic biomarkers, along with ultrasonographic evaluation and standard scores of disease activity, for a proper and complete view of the inflammatory process and therapeutic management efficiency, in order to aim for real remission/low disease activity and improve the long-term prognosis of our patients.

We acknowledge that this study conducted only in our reference center has certain limitations, due to its descriptive nature, as well as the small sample size which included only female patients. Although our data suggest some correlations between serum levels of biomarkers and disease activity, these observations would certainly benefit from a further extension of the study, with multicenter involvement. Furthermore, a future prospective study could analyze a dynamic correlation of serum biomarkers with conventional disease activity scores and also assess sensitivity to treatment. The potential insight gained from more research on this topic can establish what serum biomarkers can be included in clinical practice as additional therapeutic targets.

## 5. Conclusions

Our preliminary study showed that proangiogenic biomarkers and proinflammatory cytokines are associated with disease activity in RA. They also correlate with commonly used inflammatory markers and the degree of structural changes in the joints assessed through ultrasonography. Although we included a relatively low number of subjects from a single center, the data can certainly constitute a start point for future extensive research.

The potential insight gained from more research on this topic can establish what serum biomarkers can be included in clinical practice as additional therapeutic targets.

## Figures and Tables

**Figure 1 diagnostics-13-01653-f001:**
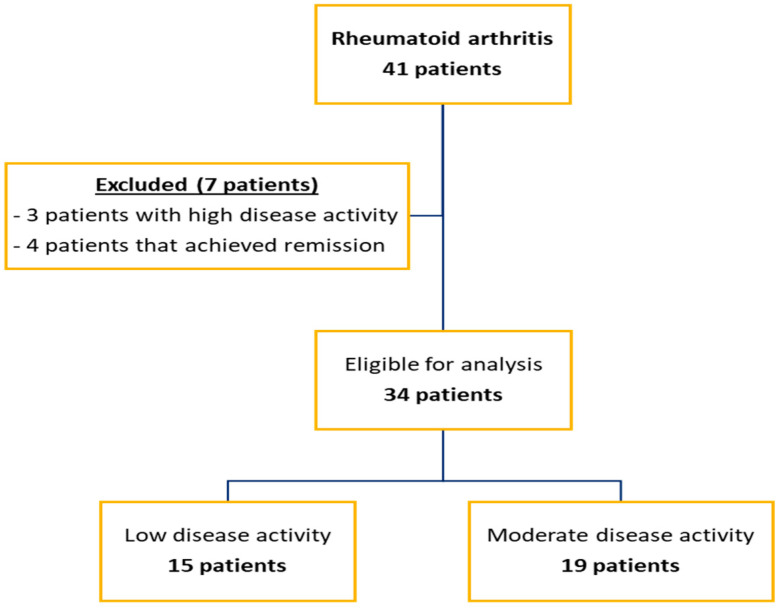
Flow chart of patient inclusion.

**Figure 2 diagnostics-13-01653-f002:**
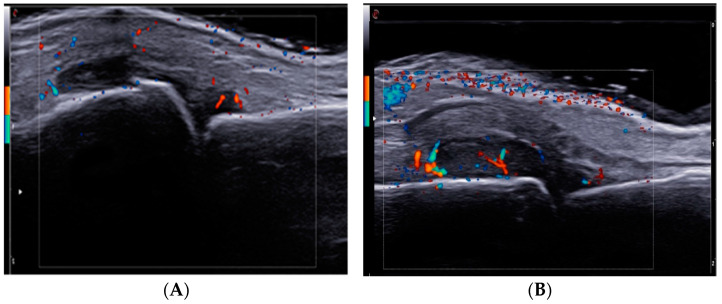
Metacarpophalangeal (MCP) joint in longitudinal view, showing a third grade GS score synovitis (**A**) and with PD signal, second grade (**B**).

**Figure 3 diagnostics-13-01653-f003:**
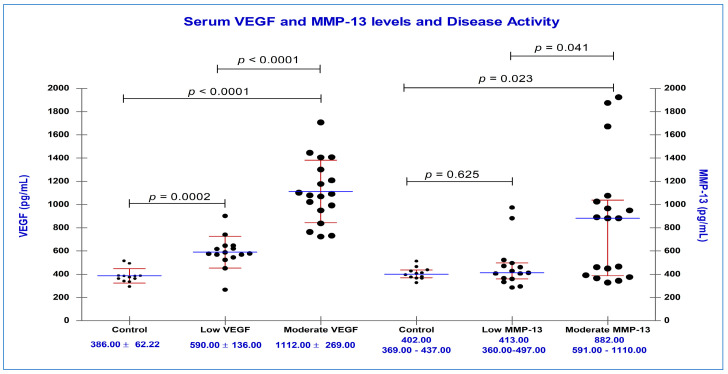
Serum concentrations of VEGF (pg/mL), MMP-13 (pg/mL) and disease activity, based on DAS28_(4v)_ CRP vs. controls; black circles represent serum concentrations of VEGF and MMP-13 from individual serum samples; blue horizontal lines represent mean (VEGF) and median (MMP-13) values, accompanied by standard deviation (VEGF) and interquartile range (MMP-13) represented by red horizontal lines.

**Figure 4 diagnostics-13-01653-f004:**
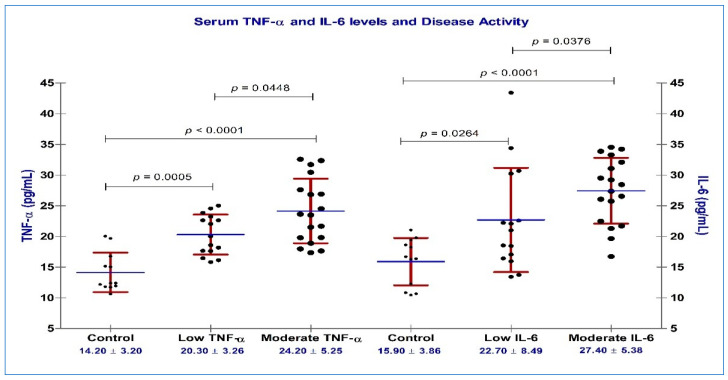
Serum concentrations of TNF-α (pg/mL) and IL-6 (pg/mL) and disease activity, based on DAS28_(4v)_ CRP vs. controls; black circles represent serum concentrations of TNF-α and IL-6 from individual serum samples; blue horizontal lines represent mean values, accompanied by standard deviation represented by red horizontal lines.

**Figure 5 diagnostics-13-01653-f005:**
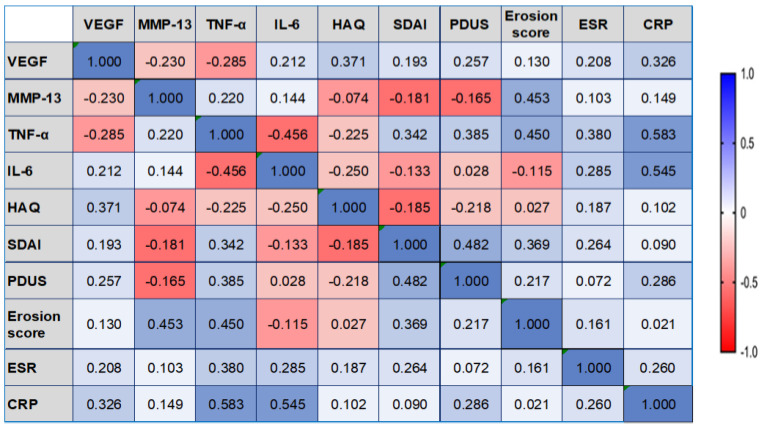
The correlation heatmap between measured indicators (colors range from bright blue for strong positive correlations to bright red for strong negative correlations), in L group. VEGF: vascular endothelial growth factor; MMP-13: matrix metalloproteinase 13; TNF-α: tumor necrosis factor-alpha; IL-6: interleukin 6; HAQ: Health Assessment Questionnaire; SDAI: Simplified Disease Activity Index; PDUS: power Doppler (PD) ultrasound (US); ESR: erythrocytes sedimentation rate; CRP: C-reactive protein.

**Figure 6 diagnostics-13-01653-f006:**
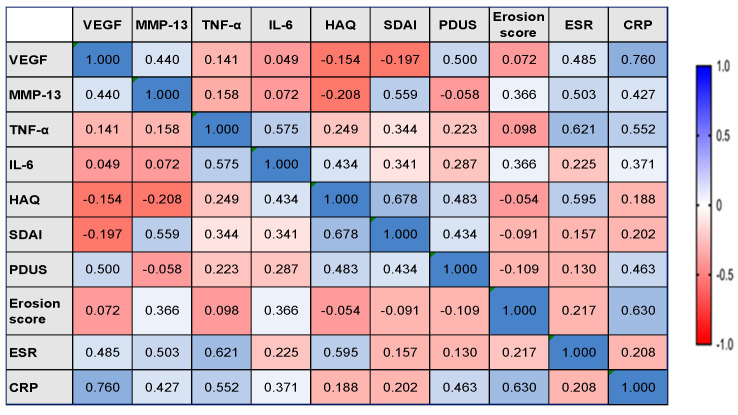
The correlation heatmap between measured indicators (colors range from bright blue for strong positive correlations to bright red for strong negative correlations), in M group. VEGF: vascular endothelial growth factor; MMP-13: matrix metalloproteinase 13; TNF-α: tumor necrosis factor-alpha; IL-6: interleukin 6; HAQ: Health Assessment Questionnaire; SDAI: Simplified Disease Activity Index; PDUS: power Doppler (PD) ultrasound (US); ESR: erythrocytes sedimentation rate; CRP: C-reactive protein.

**Figure 7 diagnostics-13-01653-f007:**
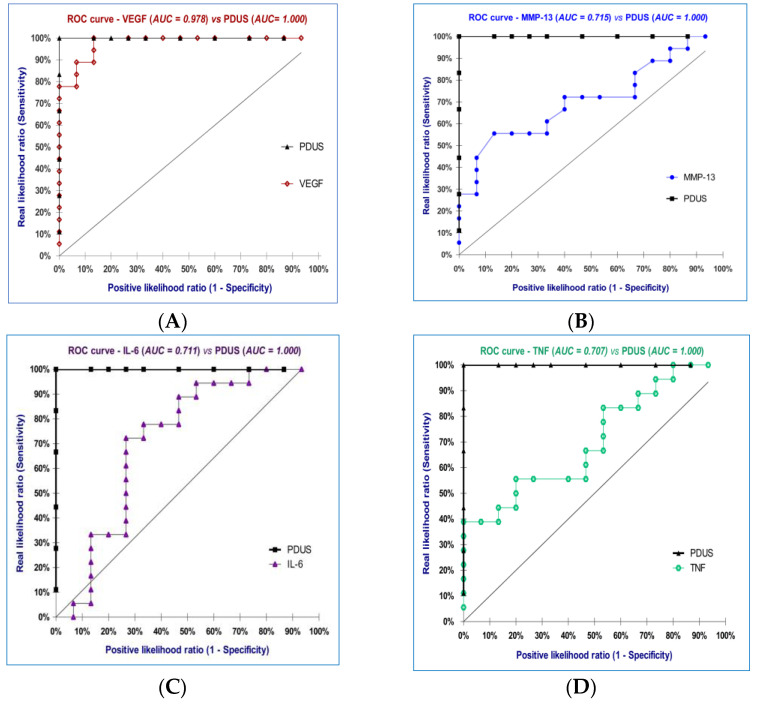
ROC curve for VEGF (**A**), MMP-13 (**B**), IL-6 (**C**), TNF (**D**), CRP (**E**) vs. PDUS.

**Table 1 diagnostics-13-01653-t001:** Demographics and clinical characteristics of the patients.

Parameter (Mean ± SD)	C (*n* = 12)	L (*n* = 15)	M (*n* = 19)	*p*-Value
Age [years]	50.36 ± 13.38	52.14 ± 10.20	54.70 ± 10.30	0.356
Urban (%)	6 (50.00%)	7 (46.67%)	7 (36.84%)	-
Rural (%)	6 (50.00%)	8 (53.33%)	12 (63.16%)	-
ESR (mm/h)	10.41 ± 4.36	53.90 ± 15.70	65.00 ± 20.10	0.054
RF (U/L)	2.54 ± 1.98	64.60 ± 39.70	103.00 ± 99.20	0.307
CRP (mg/L)	4.31 ± 1.67	31.50 ± 15.10	36.30 ± 14.80	0.322
HAQ	-	1.60 (1.10–2.10)	1.77 (1.18–2.10)	0.357
SDAI	-	28.50 ± 5.61	32.10 ± 5.85	0.175
PDUS score	-	9.60 ± 2.72	17.33 ± 1.64	<0.0001 *
Erosion score	-	2.00 ± 0.85	4.67 ± 1.64	<0.0001 *

ESR: erythrocytes sedimentation rate; RF: rheumatoid factor; CRP: C-reactive protein; DAS28_(4v)_ CRP: Disease Activity Score using 28 joints with 4 variables, using C reactive protein; HAQ: Health Assessment Questionnaire; SDAI: Simplified Disease Activity Index; PDUS: power Doppler (PD) ultrasound (US); SD: standard deviation; * statistically significant *p*-value.

**Table 2 diagnostics-13-01653-t002:** Serum biomarkers concentrations for RA patients.

Parameter	Moderate RA (n = 19)			Low RA (n = 15)		
	*Moderate*	*Control*	*p*	*Low*	*Control*	*p*
**VEGF** **(pg/mL)** ** *(Mean ± SD)* **	1112.00 ± 269.00	386.00 ± 62.20	*<0.0001*	590.00 ± 136.00	386.00 ± 62.20	*0.0002*
**MMP-13 ** **(pg/mL)** **Median ** **(Interquartile range)**	882.00 (388.00 – 1040.00)	402.00 (369.00 – 437.00)	*0.023*	413.00 (360.00 – 497.00)	402.00 (369.00 – 437.00)	0.625
**TNF-α** **(pg/mL)** ** *(Mean ± SD)* **	24.20 ± 5.25	14.20 ± 3.20	*<0.0001*	20.30 ± 3.26	14.20 ± 3.20	*0.0005*
**IL-6** **(pg/mL)** ** *(Mean ± SD)* **	27.40 ± 5.38	15.90 ± 3.86	*<0.0001*	22.70 ± 8.49	15.90 ± 3.86	*0.026*

RA: rheumatoid arthritis; VEGF: vascular endothelial growth factor; MMP-13: matrix metalloproteinase 13; TNF-α: tumor necrosis factor-alpha; IL-6: interleukin 6; SD: standard deviation; data were analyzed for statistical significance using Mann–Whitney tests between groups.

**Table 3 diagnostics-13-01653-t003:** Correlations between DAS28_(4v)_ CRP and serum concentrations of VEGF, MMP-13, TNF-α, and IL-6.

Parameter	VEGF	MMP-13	TNF	IL-6
L group (Low RA DAS28_(4v)_ CRP)	rho = 0.001 *p* = 0.998	rho = 0.580 *p* = 0.023 *	rho = −0.071 *p* = 0.801	rho = 0.477 *p* = 0.072
M group (Moderate RA DAS28_(4v)_ CRP)	rho = 0.240 *p* = 0.338	rho = 0.530 *p* = 0.024 *	rho = −0.128 *p* = 0.613	rho = 0.148 *p* = 0.558

RA: rheumatoid arthritis; VEGF: vascular endothelial growth factor; MMP-13: matrix metalloproteinase 13; TNF-α: tumor necrosis factor-alpha; IL-6: interleukin 6; CRP: C-reactive protein; DAS28_(4v)_ CRP: Disease Activity Score using 28 joints with 4 variables, using C reactive protein; 𝑟ho: Spearman correlation coefficient; * statistically significant correlations.

**Table 4 diagnostics-13-01653-t004:** Diagnostic performance of the investigated parameters.

Parameter	AUC	Threshold Values	Sensitivity %	Specificity %	Youden Index	*p*-Value
**VEGF**	0.978	926.10	77.78	100.00	0.778	<0.0001
**MMP-13**	0.715	703.10	55.56	86.67	0.412	0.039
**IL-6**	0.711	24.17	72.22	73.33	0.456	0.036
**TNF-α**	0.707	23.36	55.56	80.00	0.356	0.043
**CRP**	0.691	29.46	55.56	66.67	0.222	0.032
**PDUS**	1.000	14.50	100.00	100.00	1.000	<0.0001
**DAS28_(4v)_ CRP**	1.000	3.70	100.00	100.00	1.000	<0.0001

VEGF: vascular endothelial growth factor; MMP-13: matrix metalloproteinase 13; TNF-α: tumor necrosis factor-alpha; IL-6: interleukin 6; CRP: C-reactive protein; PDUS: power Doppler ultrasound; DAS28_(4v)_ CRP: Disease Activity Score using 28 joints with 4 variables, using C reactive protein; AUC: area under the ROC curve.

## Data Availability

The data used to support the findings of this study are available from the corresponding author upon reasonable request.

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
