# Peer review of "MMP-13, VEGF, and Disease Activity in a Cohort of Rheumatoid Arthritis Patients"

_diagnostics, 2023, doi:10.3390/diagnostics13091653_

Round 1
Reviewer 1 Report
Overall, this study presents a well-structured analysis of the association between serum concentrations of MMP-13, VEGF, TNF-α, and IL-6 and RA activity in women. However, there are several significant omissions that should be addressed.
1. While the study aimed to evaluate the diagnostic value of MMP-13, VEGF, TNF-α, and IL-6 in RA activity, the title only mentions MMP-13 and VEGF, and the Discussion section lacks comparative data on all cytokines studied.
2. The Introduction section should be supplemented with a clear statement on the research gap in this area, as the involvement of all markers studied in RA is already well-known.
3. The study aim is too lengthy and should be reformulated.
4. The Methods section should include the study design, basis for the study's sample size, and a flowchart diagram to clarify the confusion between the mentioned 41 patients in the Methods section and the 34 patients in the Results section.
5. The Results section lacks sufficient information on patient characteristics, and a footnote on *Statistically significant p-value is unnecessary given the exact p-value presented in Table 2.
6. The Discussion section would benefit from an inclusion of the study's strengths.
In conclusion, with the recommended revisions, this paper has the potential to be published.
Overall, the English language in the paper is commendable. However, it is advisable to exercise caution and ensure the appropriate usage and accuracy of prepositions.
Author Response
Dear Reviewer,
Thank you very much for taking the time to analyze our manuscript, as well as for your kind appreciation and valuable suggestions.
All the typing recommended changes were performed in the body of our manuscript, with the Track Changes function activated.
Comments and Suggestions for Authors
Overall, this study presents a well-structured analysis of the association between serum concentrations of MMP-13, VEGF, TNF-α, and IL-6 and RA activity in women. However, there are several significant omissions that should be addressed.
- While the study aimed to evaluate the diagnostic value of MMP-13, VEGF, TNF-α, and IL-6 in RA activity, the title only mentions MMP-13 and VEGF, and the Discussion section lacks comparative data on all cytokines studied.
- As we mentioned in the manuscript, in choosing the research theme, we started from the finding that there is a lack of data regarding the involvement of MMP-13 in RA, as well as the supposed association between MMP-13 and VEGF. As a result, in the Discussions section, we mentioned the results obtained in our study about MMP-13 and VEGF, to present the novelty of the study. We did not present in detail the results for TNF and IL-6 because they did not bring new information.
- The Introduction section should be supplemented with a clear statement on the research gap in this area, as the involvement of all markers studied in RA is already well-known.
- In the introduction, we specifically focused on the MMP-13 and VEGF, the main biomarkers investigated in the study, markers for which there is a research gap in this field.
- The study aim is too lengthy and should be reformulated.
- Revised
- The Methods section should include the study design, basis for the study's sample size, and a flowchart diagram to clarify the confusion between the mentioned 41 patients in the Methods section and the 34 patients in the Results section.
- Inserted
- The Results section lacks sufficient information on patient characteristics, and a footnote on *Statistically significant p-value is unnecessary given the exact p-value presented in Table 2.
- Revised Table 2
- The Discussion section would benefit from an inclusion of the study's strengths.
- We consider that in the Discussions section, we have included as study's strengths:
- lines 516-518: To our knowledge, this is the first study to evaluate the predictive utility of the MMP-13. According to our results, although MMP-13, TNF-α, and CRP have the same Sn (55.56%), MMP-13 has a better Sp (86.67%) for patients with moderate disease activity.
- lines 507-511: Comparing the ROC curves for the seven parameters analyzed (MMP-13, VEGF, TNF-α, IL-6, CRP, PDUS, DAS28(4v) CRP), we observed that the best distinguish between patients with Moderate and Low disease activity can be achieved using PDUS, followed by VEGF, MMP-13, and IL-6
- lines 416-417: Analyzing our data, we concluded that higher serum levels of MMP-13 and VEGF are present in RA patients compared to controls. Also, we identified moderately positive correlations between MMP-13, VEGF, and disease activity score.
- lines 499-504: Our results revealed high serum levels of MMP-13, directly correlated to disease activity score. We also registered statistically significant differences in MMP-13 values for patients with moderate disease activity, when compared to controls, an observation that didn’t apply to subjects with a DAS28(4v) CRP score corresponding to low disease activity. When analyzing the relationship between the MMP-13 and VEGF, a significant result was observed only in the Moderate disease activity group.
- lines 533-535: our data suggests some correlations between serum levels of biomarkers and disease activity, these observations would certainly benefit from a further extension of the study, with multicentric involvement.
Reviewer 2 Report
Boldeanu and colleagues investigate the relationships between the serum levels of VEGF, MMP-13 and disease activity in patients with RA as well as with the ultrasonographic assessment and the levels of the common cytokines TNF-a and IL-6. The authors discussed the potential for the use of these biomarkers in clinical diagnostics and as possible treatment targets. The study is well designed and the results are logically presented and correctly illustrated.
My comments are related to technical improvements of the manuscript.
1) In Tables 1 and 2, the number of individuals (N) should be given with small “n” (n=12 etc.).
2) Please, remove column 4 (last column) in Table 2, because it duplicates the data from the previous two columns, moreover Figures 2 ad 3 present the same results.
3) Lines 447-449. “One very important correlation of our study, confirmed by other reports, was registered comparing the serum levels of VEGF and PDUS score, for patients with a moderate 448 disease activity score.” There is no way results from this study (not published yet) to be confirmed by other articles. Your results can confirm other reports. Please, rephrase this sentence and provide (if necessary) references.
4) Line 454: GSUS is used for the first time here, so, please give the full name.
5) Line 693. Please, delete the empty brackets ().
Please, change the words "multicenter" study (line 422), and "single-center" study (line 531). Line 429 (evolution) ???
Author Response
Dear Reviewer,
Thank you very much for taking the time to analyze our manuscript, as well as for your kind appreciation and valuable suggestions.
All the typing recommended changes were performed in the body of our manuscript, with the Track Changes function activated.
Comments and Suggestions for Authors
Boldeanu and colleagues investigate the relationships between the serum levels of VEGF, MMP-13 and disease activity in patients with RA as well as with the ultrasonographic assessment and the levels of the common cytokines TNF-a and IL-6. The authors discussed the potential for the use of these biomarkers in clinical diagnostics and as possible treatment targets. The study is well designed and the results are logically presented and correctly illustrated.
My comments are related to technical improvements of the manuscript.
1) In Tables 1 and 2, the number of individuals (N) should be given with small “n” (n=12 etc.).
- Revised
2) Please, remove column 4 (last column) in Table 2, because it duplicates the data from the previous two columns, moreover Figures 2 ad 3 present the same results.
- Revised
3) Lines 447-449. “One very important correlation of our study, confirmed by other reports, was registered comparing the serum levels of VEGF and PDUS score, for patients with a moderate 448 disease activity score.” There is no way results from this study (not published yet) to be confirmed by other articles. Your results can confirm other reports. Please, rephrase this sentence and provide (if necessary) references.
- Revised
4) Line 454: GSUS is used for the first time here, so, please give the full name.
- Revised
5) Line 693. Please, delete the empty brackets ().
- Revised
Comments on the Quality of English Language
Please, change the words "multicenter" study (line 422), and "single-center" study (line 531). Line 429 (evolution) ???
- Revised